# CONDITIONAL NETWORK EMBEDDINGS

**Bo Kang, Jefrey Lijffijt & Tijl De Bie**
Department of Electronics and Information Systems (ELIS), IDLab
Ghent University
Ghent, Belgium
`{firstname.lastname}@ugent.be`

## ABSTRACT

Network Embeddings (NEs) map the nodes of a given network into $d$-dimensional Euclidean space $\mathbb{R}^d$. Ideally, this mapping is such that 'similar' nodes are mapped onto nearby points, such that the NE can be used for purposes such as link prediction (if 'similar' means being 'more likely to be connected' or 'having similar neighborhoods') or classification (if 'similar' means 'being more likely to have the same label'). In recent years various methods for NE have been introduced, all following a similar strategy: defining a notion of similarity between nodes, a distance measure in the embedding space, and a loss function that penalizes large distances for similar nodes and small distances for dissimilar nodes.

A difficulty faced by existing methods is that certain networks are fundamentally hard to embed due to their structural properties: (approximate) multipartiteness, certain degree distributions, assortativity, etc. To overcome this, we introduce a conceptual innovation to the NE literature and propose to create *Conditional Network Embeddings* (CNEs); embeddings that maximally add information with respect to given structural properties (e.g. node degrees, block densities, etc.). We use a simple Bayesian approach to achieve this, and propose a block stochastic gradient descent algorithm for fitting it efficiently. We demonstrate that CNEs are superior for link prediction and multi-label classification when compared to state-of-the-art methods, and this without adding significant mathematical or computational complexity. Finally, we illustrate the potential of CNE for network visualization.

## 1 INTRODUCTION

Network Embeddings (NEs) map nodes into $d$-dimensional Euclidean space $\mathbb{R}^d$ such that an ordinary distance measure allows for meaningful comparisons between nodes. Embeddings directly enable the use of a variety of machine learning methods (classification, clustering, etc.) on networks, explaining their exploding popularity. NE approaches typically have three components (Hamilton et al., 2017): (1) A measure of similarity between nodes. E.g. nodes can be deemed more similar if they are adjacent, have strongly overlapping neighborhoods, or are otherwise close to each other (link and path-based measures) (Grover & Leskovec, 2016; Perozzi et al., 2014; Tang et al., 2015), or if they have similar functional properties (structural measures) (Ribeiro et al., 2017). (2) A metric in the embedding space. (3) A loss function comparing similarity between node pairs in the network with the proximity of their embeddings. A good NE is then one for which the average loss is small.

**Limitations of existing NE approaches**   A problem with all NE approaches is that networks are fundamentally more expressive than embeddings in Euclidean spaces. Consider for example a bipartite network $G = (V, U, E)$ with $V, U$ two disjoint sets of nodes and $E \subseteq V \times U$ the set of links. It is in general impossible to find an embedding in $\mathbb{R}^d$ such that $v \in V$ and $u \in U$ are close for all $(v, u) \in E$, while all pairs $v, v' \in V$ are far from each other, as well as all pairs $u, u' \in U$. To a lesser extent, this problem will persist in approximately bipartite networks, or more generally (approximately) $k$-partite networks such as networks derived from stochastic block models.[1] This

---

[1]For example multi-relational data can be represented as a $k$-partite network, where the schema specifies between which types of objects links may exist. Another example is a heterogeneous information network, where no schema is provided but links are more or less common depending on the (specified) types of the nodes.

shows that first-order similarity (i.e. adjacency) in networks cannot be modeled well using a NE. Similar difficulties exist for second-order proximity (i.e. neighborhood overlap) and other node similarity notions. A more subtle example is a network with a power law degree distribution. A first-order similarity NE will tend to embed high degree nodes towards the center (to be close to lots of other nodes), while the low degree nodes will be on the periphery. Yet, this effect reduces the embedding's degrees of freedom for representing similarity independent of node degree.

**CNE: the idea** To address these limitations of NEs, we propose a principled probabilistic approach— dubbed *Conditional Network Embedding (CNE)*—that allows optimizing embeddings w.r.t. certain prior knowledge about the network, formalized as a prior distribution over the links. This prior knowledge may be derived from the network itself such that no external information is required.

A combined representation of a prior based on structural information and a Euclidean embedding makes it possible to overcome the problems highlighted in the examples above. For example, nodes in different blocks of an approximately $k$-partite network need not be particularly distant from each other if they are a priori known to belong to the same block (and hence are unlikely or impossible to be connected a priori). Similarly, high degree nodes need not be embedded near the center of the point cloud if they are known to have high degree, as it is then known that they are connected to many other nodes. The embedding can thus focus on encoding which nodes in particular it is connected to.

CNE is also potentially useful for network visualization, with the ability to filter out certain information by using it as a prior. For example, suppose the nodes in a network represent people working in a company with a matrix-structure (vertical being units or departments, horizontal contents such as projects) and links represent whether they interact a lot. If we know the vertical structure, we can construct an embedding where the prior is the vertical structure. The information that the embedding will try to capture corresponds to the horizontal structure. The embedding can then be used in downstream analysis, e.g., to discover clusters that correspond to teams in the horizontal structure.

**Contributions and outline** Our contributions can be summarized as follows:

- This paper introduces the *concept of NE conditional on certain prior knowledge* about the network.
- Section 2 presents *CNE ('Conditional Network Embedding')*, which realizes this idea by using Bayes rule to combine a prior distribution for the network with a probabilistic model for the Euclidean embedding conditioned on the network. This yields the posterior probability for the network conditioned on the embedding, which can be maximized to yield a maximum likelihood embedding. Section 2.2 describes *a scalable algorithm* based on block stochastic gradient descent.
- Section 3 reports on *extensive experiments*, comparing with state-of-the-art baselines on link prediction and multi-label classification, on commonly used benchmark networks. These experiments show that CNE's link prediction accuracy is consistently superior. For multi-label classification CNE is consistently best on the Macro-$F_1$ score and best or second best on the Micro-$F_1$ score. These results are achieved with *considerably lower-dimensional embeddings* than the baselines. A case study also demonstrates the usefulness of CNE in *exploratory data analysis* of networks.
- Section 4 gives a brief overview of *related work*, before *concluding* the paper in Section 5.
- All code, including code for repeating the experiments, and links to the datasets are available at: `https://bitbucket.org/ghentdatascience/cne`.

## 2 METHODS

Section 2.1 introduces the probabilistic model used by CNE, and Section 2.2 describes an algorithm for optimizing it to find an optimal CNE. Before doing that, let us introduce some notation. An undirected network is denoted $G = (V, E)$ where $V$ is a set of $n = |V|$ nodes and $E \subseteq \binom{V}{2}$ is the set of links (also known as edges). A link is denoted by an unordered node pair $\{i, j\} \in E$. Let $\hat{\mathbf{A}}$ denote the network's adjacency matrix, with element $\hat{a}_{ij} = 1$ for $\{i, j\} \in E$ and $\hat{a}_{ij} = 0$ otherwise. The goal of NE (and thus of CNE) is to find a mapping $f : V \to \mathbb{R}^d$ from nodes to $d$-dimensional real vectors. The resulting embedding is denoted $\mathbf{X} = (\mathbf{x}_1, \mathbf{x}_2, \ldots, \mathbf{x}_n)' \in \mathbb{R}^{n \times d}$.

## 2.1 THE CONDITIONAL NETWORK EMBEDDING MODEL

The newly proposed method CNE aims to find an embedding $\mathbf{X}$ that is maximally informative about the given network $G$, formalized as a Maximum Likelihood (ML) estimation problem:

$$\underset{\mathbf{X}}{\text{argmax}} \;\; P(G|\mathbf{X}). \tag{1}$$

Innovative about CNE is that we do not postulate the likelihood function $P(G|\mathbf{X})$ directly, as is common in ML estimation. Instead, we use a generic approach to derive prior distributions for the network $P(G)$, and we postulate the density function for the data conditional on the network $p(\mathbf{X}|G)$. This allows one to introduce any prior knowledge about the network into the formulation, through a simple application of Bayes rule[2]: $P(G|\mathbf{X}) = \frac{p(\mathbf{X}|G)P(G)}{p(\mathbf{X})}$. The consequence is that the embedding will not need to represent any information that is already represented by the prior $P(G)$.

Section 2.1.1 describes how a broad class of prior information types can be modeled for use by CNE. Section 2.1.2 describes a possible conditional distribution (albeit an improper one), the one we used for the particular CNE method in this paper. Section 2.1.3 describes the posterior distribution.

### 2.1.1 THE PRIOR DISTRIBUTION FOR THE NETWORK

We wish to be able to model a broad class of prior knowledge types in the form of a manageable prior probability distribution $P(G)$ for the network. Let us first focus on three common types of prior knowledge: knowledge about the overall network density, knowledge about the individual node degrees, and knowledge about the edge density within or between particular subsets of the nodes (e.g. for multipartite networks). Each of these can be expressed as sets of constraints on the expectations of the sum of various subsets $S \subseteq \binom{V}{2}$ of elements from the adjacency matrix: $\mathbb{E}\left\{\sum_{\{i,j\}\in S} a_{ij}\right\} = \sum_{\{i,j\}\in S} \hat{a}_{ij}$, where the expectation is taken w.r.t. the sought prior distribution $P(G)$. In the 1st case, $S = \binom{V}{2}$; in the 2nd case, $S = \{(i,j)|j \in V, j \neq i\}$ for information on the degree of node $i$; and in the 3rd case $S = \{(i,j)|i \in A, j \in B, i \neq j\}$ for specified sets $A, B \in V$.

Such constraints do not determine $P(G)$ fully, so we determine $P(G)$ as the distribution with maximum entropy from all distributions satisfying all these constraints. Adriaens et al. (2017); van Leeuwen et al. (2016) showed that finding this distribution is a convex optimization problem that can be solved efficiently, particularly for sparse networks. They also showed that the resulting distribution is a product of independent Bernoulli distributions, one for each element of the adjacency matrix:

$$P(G) = \prod_{\{i,j\}\in\binom{V}{2}} P_{ij}^{\hat{a}_{ij}}(1 - P_{ij})^{1-\hat{a}_{ij}}, \tag{2}$$

where $P_{ij} \in [0, 1]$ is the probability that $\{i, j\}$ is linked in the network under this distribution. They showed that all these $P_{ij}$ can be expressed in terms of a limited number of parameters, namely the unique Lagrange multipliers for the prior knowledge constraints in the maximum entropy problem. In practice, the number of such unique Lagrange multipliers is far smaller than $n$.

The three cases discussed above are merely examples of how constraints on the expectation of subsets of the elements of the adjacency matrix can be useful in practice. For example, if nodes are ordered in some way (e.g. according to time), it could be used to express the fact that nodes are connected only to nodes that are not too distant in that ordering. Moreover, the above results continue to hold for constraints that are on *weighted* linear combinations of elements of the adjacency matrix. This makes it possible to express other kinds of prior knowledge, e.g. on the relation between connectedness and distance in a node order (if provided), or on the network's (degree) assortativity. A detailed discussion and empirical analysis of such alternatives is deferred to further work.

### 2.1.2 THE DISTRIBUTION OF THE DATA CONDITIONED ON THE NETWORK

We now move on to postulating the conditional density $P(\mathbf{X}|G)$. Clearly, any rotation or translation of an embedding should be considered equally good, as we are only interested in distances between pairs

---

[2]Note that this approach is uncommon: despite the usage of Bayes rule, it is not Maximum A Posteriori (MAP) estimation as the chosen embedding $\mathbf{X}$ is the one maximizing the likelihood of the network.

of nodes in the embedding. Thus, the pairwise distances between points, denoted as $d_{ij} \triangleq \|\mathbf{x}_i - \mathbf{x}_j\|_2$ for points $\mathbf{x}_i, \mathbf{x}_j \in \mathbb{R}^d$, must form a set of sufficient statistics.

The density should also reflect the fact that connected node pairs tend to be embedded to nearby points, while disconnected node pairs tend to be embedded to more distant points. Let us focus initially on the marginal density of $d_{ij}$ conditioned on $G$. The proposed model assumes that given $\hat{a}_{ij}$ (i.e. knowledge of whether $\{i, j\} \in E$ or not), $d_{ij}$ is conditionally independent of the rest of the adjacency matrix. More specifically, we model the conditional distribution for the distances $d_{ij}$ given $\{i, j\} \in E$ as half-normal $\mathcal{N}_+$ (Leone et al., 1961) with spread parameter $\sigma_1 > 0$:[3]

$$p\left(d_{ij}|\{i, j\} \in E\right) = \mathcal{N}_+\left(d_{ij}|\sigma_1^2\right), \tag{3}$$

and the distribution of distances $d_{kl}$ with $\{k, l\} \notin E$ as half-normal with spread parameter $\sigma_2 > \sigma_1$:

$$p\left(d_{kl}|\{k, l\} \notin E\right) = \mathcal{N}_+\left(d_{kl}|\sigma_2^2\right). \tag{4}$$

The choice of $0 < \sigma_1 < \sigma_2$ will ensure the embedding reflects the neighborhood proximity of the network. Indeed, the differences between the embedded nodes that are not connected in the network are expected to be larger than the differences between the embedding of connected nodes. Without losing generality (as it merely fixes the scale), we set $\sigma_1 = 1$ through out this paper.

It is clear that the distances $d_{ij}$ cannot be independent of each other (e.g. the triangle inequality entails a restriction of the range of $d_{ij}$ given the values of $d_{ik}$ and $d_{jk}$ for some $k$). Nevertheless, akin to Naive Bayes, we still model the joint distribution of all distances (and thus of the embedding $\mathbf{X}$ up to a rotation/translation) as the product of the marginal densities for all pairwise distances:

$$p(\mathbf{X}|G) = \prod_{\{i,j\} \in E} \mathcal{N}_+\left(d_{ij}|\sigma_1^2\right) \cdot \prod_{\{k,l\} \notin E} \mathcal{N}_+\left(d_{kl}|\sigma_2^2\right). \tag{5}$$

This is an improper density function, due to the constraints imposed by Euclidean geometry. Indeed, certain combinations of pairwise distances should be assigned a probability 0 as they are geometrically impossible. As a result, $p(\mathbf{X}|G)$ is also not properly normalized. Yet, even though $p(\mathbf{X}|G)$ is improper, it can still be used to derive a properly normalized posterior for $G$ as detailed next.

### 2.1.3 THE POSTERIOR OF THE NETWORK CONDITIONED ON THE EMBEDDING

The (also improper) marginal density $p(\mathbf{X})$ can now be computed as:

$$p(\mathbf{X}) = \sum_G p(\mathbf{X}|G)P(G) = \sum_G \prod_{\{i,j\} \in E} \mathcal{N}_+\left(d_{ij}|\sigma_1^2\right) P_{ij} \cdot \prod_{\{k,l\} \notin E} \mathcal{N}_+\left(d_{kl}|\sigma_2^2\right) (1 - P_{kl}),$$

$$= \prod_{i,j} \left[\mathcal{N}_+\left(d_{ij}|\sigma_1^2\right) P_{ij} + \mathcal{N}_+\left(d_{ij}|\sigma_2^2\right) (1 - P_{ij})\right].$$

We now have all ingredients to compute the posterior of the network conditioned on the embedding by a simple application of Bayes' rule:

$$P(G|\mathbf{X}) = \frac{p(\mathbf{X}|G) \cdot P(G)}{p(\mathbf{X})} = \prod_{\{i,j\} \in E} \frac{\mathcal{N}_+\left(d_{ij}|\sigma_1^2\right) P_{ij}}{\mathcal{N}_+\left(d_{ij}|\sigma_1^2\right) P_{ij} + \mathcal{N}_+\left(d_{ij}|\sigma_2^2\right) (1 - P_{ij})}$$

$$\cdot \prod_{\{k,l\} \notin E} \frac{\mathcal{N}_+\left(d_{kl}|\sigma_2^2\right) (1 - P_{kl})}{\mathcal{N}_+\left(d_{kl}|\sigma_1^2\right) P_{kl} + \mathcal{N}_+\left(d_{kl}|\sigma_2^2\right) (1 - P_{kl})}. \tag{6}$$

This is the likelihood function to be maximized in order to get the ML embedding. Note that, although it was derived using the improper density function $p(\mathbf{X}|G)$, thanks to the normalization with the (equally improper) $p(\mathbf{X})$, this is indeed a properly normalized distribution.

---

[3]A half-normal distribution, with density denoted here as $\mathcal{N}_+(\cdot|\sigma^2)$, is a zero-mean normal distribution with standard deviation $\sigma$, conditioned on the random variable being positive. Of course the standard deviation of the conditioned normal distribution is not equal to $\sigma$, so we refer to $\sigma$ more loosely as its spread parameter.

## 2.2 Finding the most informative embedding

Maximizing the likelihood function $P(G|\mathbf{X})$ is a non-convex optimization problem. We propose to solve it using a block stochastic gradient descent approach, explained below. The gradient of the likelihood function (Eq. 6) with respect to the embedding $\mathbf{x}_i$ of node $i$ is:[4]

$$\nabla_{\mathbf{x}_i} \log\left(P(G|\mathbf{X})\right) = 2 \sum_{j:\{i,j\}\in E} (\mathbf{x}_i - \mathbf{x}_j) P(a_{ij} = 0|\mathbf{X}) \left(\frac{1}{\sigma_2^2} - \frac{1}{\sigma_1^2}\right)$$

$$+ 2 \sum_{j:\{i,j\}\notin E} (\mathbf{x}_i - \mathbf{x}_j) P(a_{ij} = 1|\mathbf{X}) \left(\frac{1}{\sigma_1^2} - \frac{1}{\sigma_2^2}\right). \tag{7}$$

As $\left(\frac{1}{\sigma_2^2} - \frac{1}{\sigma_1^2}\right) < 0$, the first summation pulls the embedding of node $i$ towards embeddings of the nodes it is connected to in $G$. Moreover, if the current prediction of the link $P(a_{ij} = 1|\mathbf{X})$ is small (i.e., if $P(a_{ij} = 0|\mathbf{X})$ is large), the pulling effect will be larger. Similarly, the second summation pushes $\mathbf{x}_i$ away from the embeddings of unconnected nodes, and more strongly so if the current prediction of a link between these two unconnected nodes $P(a_{ij} = 1|\mathbf{X})$ is larger. The magnitudes of the gradient terms are also affected by parameter $\sigma_2$ and prior $P(G)$: a large $\sigma_2$ gives stronger push and pulling effect. In our quantitative experiments we always set $\sigma_2 = 2$.

Computing this gradient w.r.t. a particular node's embedding requires computing the pairwise differences between $n$ proposed $d$-dim embedding vectors, with time complexity $\mathcal{O}(n^2 d)$ and space complexity $\mathcal{O}(nd)$. This is computationally demanding for mainstream hardware even for networks of sizes of the order $n = 1000$ and dimensionalities of the order $d = 10$, and prohibitive beyond that. To address this issue, we approximate both summations in the objective by sampling $k < n/2$ terms from each. This amounts to uniformly sampling $k$ nodes from the set of connected nodes (where $a_{ij} = 1$), and $k$ from the set of unconnected nodes (where $a_{ij} = 0$).[5] This reduces the time complexity to $\mathcal{O}(ndk)$.

Note that each of the terms is bound in norm by the diameter of the embedding, as the other factors are bound by 1 for $\sigma_1 = 1, \sigma_1 < \sigma_2$. If the diameter were bounded, a simple application of Hoeffding's inequality would demonstrate that this average is sharply concentrated around its expectation, and is thus a suitable approximation. Although there is no prior bound that holds with guarantee on the diameter of the embedding, this does shed some light on why this approach works well in practice. The choice of $k$ will in practice be motivated by computational constraints. In our experiments we set it equal or similar to the largest degree, such that the first term is computed exactly.

## 3 Experiments

We first evaluate the network representation obtained by CNE on downstream tasks typically used for evaluating NE methods: link prediction for links and multi-label classification for nodes. Then, we illustrate how to use CNE to visually explore multi-relational data.

### 3.1 Experiment setup

For the quantitative evaluations, we compare CNE against a panel of state-of-the-art baselines for NE: Deepwalk (Perozzi et al., 2014), LINE (Tang et al., 2015), node2vec (Grover & Leskovec, 2016), metapath2vec++ (Dong et al., 2017), and struc2vec (Ribeiro et al., 2017). Table 1 lists the networks used in the experiments. A brief discussion of the methods and the networks is given in the supplement.

For all methods we used their default parameter settings reported in the original papers and with d = 128. For node2vec, the hyperparameters $p$ and $q$ are tuned over a grid $p, q \in \{0.25, 0.05, 1, 2, 4\}$ using 10-fold cross validation. We repeat our experiments for 10 times with different random seeds. The final scores are averaged over the 10 repetitions.

---

[4]We refer the reader to the supplementary material for detailed derivations.

[5]If a node $i$ has a degree smaller than $k$, we sample more non-connected neighbors to make sure that $2k$ points are used for the approximation of the gradient – and conversely if a node has a degree larger than $n - k$.

Table 1: Networks used in experiments.

| Data | Type | #Nodes | #Links | #Labels |
|------|------|--------|--------|---------|
| Facebook (Leskovec & Krevl, 2015) | Friendship | 4,039 | 88,234 | – |
| arXiv ASTRO-PH (Leskovec & Krevl, 2015) | Co-authorship | 18,722 | 198,110 | – |
| Gowalla (Cho et al., 2011) | Friendship | 196,591 | 950,327 | – |
| StudentDB (Goethals et al., 2010) | Relational/k-partite | 403 | 3,429 | – |
| BlogCatalog (Zafarani & Liu, 2009) | Bloggers | 10,312 | 333,983 | 39 |
| Protein-Protein Int. (Breitkreutz et al., 2007) | Biological | 3,890 | 76,584 | 50 |
| Wikipedia (Mahoney, 2011) | Word co-occurrence | 4,777 | 184,812 | 40 |

Table 2: The AUC scores for link prediction. TimeOut means aborted after 24 hours.

| Algorithm | Facebook | PPI | arXiv | BlogCat. | Wikiped. | studentdb | Gowalla |
|-----------|----------|-----|-------|----------|----------|-----------|---------|
| Common Neigh. | 0.9735 | 0.7693 | 0.9422 | 0.9215 | 0.8392 | 0.4160 | 0.7769 |
| Jaccard Sim. | 0.9705 | 0.7580 | 0.9422 | 0.7844 | 0.5048 | 0.4160 | 0.7519 |
| Adamic Adar | 0.9751 | 0.7719 | 0.9427 | 0.9268 | 0.8634 | 0.4160 | 0.7719 |
| Prefer. Attach. | 0.8295 | 0.8892 | 0.8640 | 0.9519 | 0.9130 | 0.9106 | 0.5626 |
| Deepwalk | 0.9798 | 0.6365 | 0.9207 | 0.6077 | 0.5563 | 0.7644 | 0.7156 |
| LINE | 0.9525 | 0.7462 | 0.9771 | 0.7563 | 0.7077 | 0.8562 | 0.8173 |
| node2vec | 0.9881 | 0.6802 | 0.9721 | 0.7332 | 0.6720 | 0.8261 | 0.7984 |
| metapath2vec++ | 0.7408 | 0.8516 | 0.8258 | 0.9125 | 0.8334 | 0.9244 | 0.7769 |
| struc2vec | 0.6909 | 0.7752 | 0.7182 | 0.8631 | 0.8062 | 0.6290 | TimeOut |
| CNE (uniform) | 0.9905 | 0.8908 | 0.9865 | 0.9190 | 0.8417 | 0.9300 | 0.9738 |
| CNE (degree) | **0.9909** | **0.9115** | **0.9882** | **0.9636** | **0.9158** | **0.9439** | **0.9818** |
| CNE (block) | NA | NA | NA | NA | NA | **0.9830** | NA |

## 3.2 LINK PREDICTION

In link prediction, we randomly remove 50% of the links of the network while keeping it connected. The remaining network is thus used for training the embedding, while the removed links (positive links, labeled 1) are used as a part of the test set. Then, the test set is topped up by an equal number of negative links (labeled 0) randomly drawn from the original network. In each repetition of the experiment, the node indices are shuffled so as to obtain different train-test splits.

We compare CNE with other methods based on the area under the ROC curve (AUC). The methods are evaluated against all datasets mentioned in the previous section. CNE typically works well with small dimensionality $d$ and sample size $k$. In this experiment we set $d = 8$ and $k = 50$. Only for the two largest networks (arXiv and Gowalla), we increase the dimensionality to d = 16 to reduce underfitting. To calculate AUC, we first compute the posterior $P(a_{ij} = 1 | \mathbf{X}_{\text{train}})$ of the test links based on the embedding $\mathbf{X}_{\text{train}}$ learned on the training network. Then the AUC score is computed by comparing the posterior probability of the test links and their true labels.

In this task we first compare CNE against four simple baselines (Grover & Leskovec, 2016): Common Neighbors ($|\mathbf{N}(i) \cap \mathbf{N}(j)|$), Jaccard Similarity ($\frac{|\mathbf{N}(i) \cap \mathbf{N}(j)|}{|\mathbf{N}(i) \cup \mathbf{N}(j)|}$), Adamic-Adar Score ($\sum_{t \in \mathbf{N}(i) \cap \mathbf{N}(j)} \frac{1}{\log |\mathbf{N}(t)|}$), and Preferential Attachment ($|\mathbf{N}(i)| \cdot |\mathbf{N}(j)|$). These baselines are neighborhood based node similarity measures. We first compute pairwise similarity on the training network. Then from the computed similarities we obtain scores for testing links as the similarity between the two ending nodes. Those scores are then used to compute the AUC against the true labels.

For the NE baselines, we perform link prediction using logistic regression based on the link representation derived from the node embedding $\mathbf{X}_{\text{train}}$. The link representation is computed by applying the Hadamard operator (element wise multiplication) on the node representation $\mathbf{x}_i$ and $\mathbf{x}_j$, which is reported to give good results (Grover & Leskovec, 2016). Then the AUC score is computed by comparing the link probability (from logistic regression) of the test links with their true labels.

Table 3: The $F_1$ scores for multi-label classification.

| Algorithm | BlogCatalog | | PPI | | Wikipedia | |
|---|---|---|---|---|---|---|
| | Macro-$F_1$ | Micro-$F_1$ | Macro-$F_1$ | Micro-$F_1$ | Macro-$F_1$ | Micro-$F_1$ |
| Deepwalk | 0.2544 | 0.3950 | 0.1795 | 0.2248 | 0.1872 | 0.4661 |
| LINE | 0.1495 | 0.2947 | 0.1547 | 0.2047 | 0.1721 | **0.5193** |
| node2vec | 0.2364 | 0.3880 | 0.1844 | 0.2353 | 0.1985 | 0.4746 |
| metapath2vec++ | 0.0351 | 0.1684 | 0.0337 | 0.0726 | 0.1031 | 0.3942 |
| struc2vec | 0.0493 | 0.1653 | 0.0669 | 0.0971 | 0.1124 | 0.4019 |
| CNE-LR (degree) | 0.1833 | 0.3376 | 0.1484 | 0.1952 | 0.1370 | 0.4339 |
| CNE-LP (block+degree) | **0.2935** | **0.4002** | **0.2639** | **0.2519** | **0.3374** | 0.4839 |

**Results** The link prediction results are shown in Table 2. Even with a uniform prior (i.e. prior knowledge only on the overall density), CNE performs better than all baselines on 5 of the 7 networks. With a degree prior, however, CNE outperforms all baselines on all networks. We attribute this to the fact that the degree prior encodes information which is hard to encode using a metric embedding alone. For the multi-relational dataset studentdb, metapath2vec++, which is designed for heterogeneous data, outperforms other baselines but not CNE (regardless of the prior information). Moreover, CNE is capable of encoding the knowledge of the block structure of this multi-relational network as a prior, with each block corresponding to one node type. Doing this improves the AUC further by $3.91\%$ versus CNE with degree prior (from $94.39\%$ to $98.30\%$; i.e., a $70\%$ reduction in error).

In terms of runtime, over the seven datasets CNE is fastest in two cases, 12% slower than the fastest (metapath2vec++) in one case, and takes approximately twice as long in the four other cases (also metapath2vec++). Detailed runtime results can be found in the supplementary material.

## 3.3 MULTI-LABEL CLASSIFICATION

We performed multi-label classification on the following networks: BlogCatalog, PPI, and Wikipedia. Detailed results are given in the supplement, while Table 3 contains an excerpt of the results. All baselines are evaluated in a standard logistic regression (LR) setup (Perozzi et al., 2014).

When using logistic regression also on the CNE embeddings, CNE performs on-par, but not particularly well (row CNE-LR). This should not be a surprise though, as potentially relevant information encoded by the prior (the degrees) will not be reflected in the embedding. However, multi-label classification can easily be cast as a link prediction problem, by adding to the network a node for each label, with a link to each node to which the label applies. Predicting a label for a node then amounts to predicting a link to that label node. To evaluate this strategy, we train an embedding on the original network plus half the label links, while the other half of the label links is held out for testing.

For the baselines, this link prediction setup does not lead to consistent improvements (see supplement), but for CNE it does (row CNE-LP, where LP stands for Link Prediction, in Table 3). On Micro-$F_1$ it is best or once close second best (after LINE with LR, see Table 3), and on Macro-$F_1$ it greatly outperforms any other method, suggesting improved performance mainly on the less frequent labels.

## 3.4 VISUAL EXPLORATION OF MULTI-RELATIONAL DATA

Here we qualitatively evaluate CNE's ability to facilitate visual exploration of multi-relational data, and how a suitable choice of the prior can help with this. To this end, we use CNE to embed the studentdb dataset directly into 2-dimensional space. As a larger $\sigma_2$ in general appears to give better visual separation between node clusters, we set $\sigma_2 = 15$.

For comparison, we first apply CNE with uniform prior (overall network density). The resulting embedding (Fig. 1a) clearly separates bachelor student/courses/program nodes (upper) from the master's nodes (lower). Also observe that the embedding is strongly affected by the node degrees (coded as marker size = log degree): high degree nodes flock together in the center. E.g., these are students who interact with many other smaller degree nodes (courses/programs). Although there are no direct links between program nodes (green) and course nodes (blue), the students (red) that connect them are pulling courses towards the corresponding program and pushing away other courses.

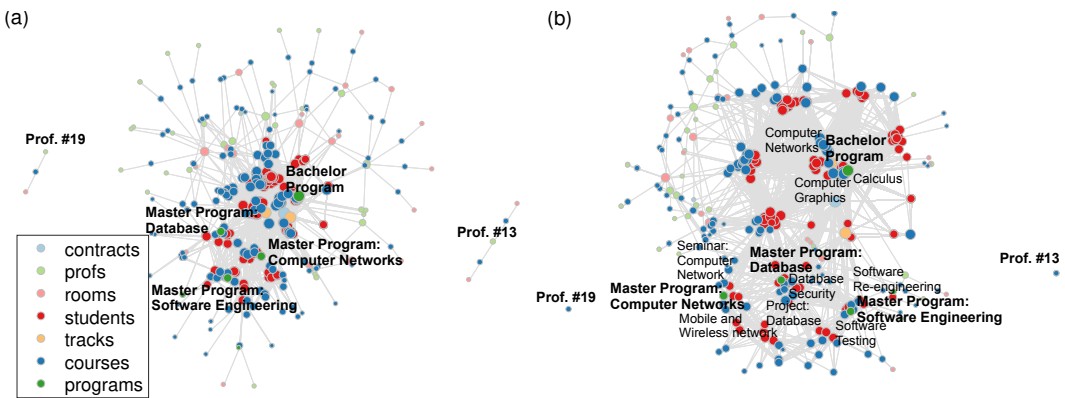

Figure 1: (a) 2-d embedding with uniform prior. (b) 2-d embedding with degree prior.

Next, we encode the individual node degrees as prior. As in this case the degree information is known, the embedding in addition shows the courses grouped around different programs, e.g.: "Bachelor Program" is close to course "Calculus"; "Master Program Computer Network" is close to course "Seminar Computer Network"; "Master Program Database" is close to course "Database Security"; "Master Program Software Engineering" is close to courses "Software Testing".

Thus, although this last evaluation remains qualitative and preliminary, it confirms that CNE with a suitable prior can create embeddings that clearly convey information in addition to the given prior.

## 4 RELATED WORK

NE methods typically have three components (Hamilton et al., 2017): (1) A similarity measure between nodes, (2) A metric in embedding space, (3) A loss function comparing proximity between nodes in embedding space with the similarity in the network. Early NE methods such as Laplacian Eigenmaps (Belkin & Niyogi, 2002), Graph factorization (Ahmed et al., 2013), GraRep (Cao et al., 2015), and HOPE (Ou et al., 2016) optimize mean-squared-error loss between Euclidean distance or inner product based proximity and link based (adjacency matrix) similarity in the network. Recently, a few NE methods define node similarity based on paths. Those paths are generated using either the adjacency matrix (LINE, Tang et al., 2015) or random walks (Deepwalk, Perozzi et al. 2014, node2vec, Grover & Leskovec 2016, methapath2vec++, Dong et al. 2017, and struc2vec Ribeiro et al. 2017). Path based embedding methods typically use inner products as proximity measure in the embedding space and optimize a cross-entropy loss. The recent struc2vec method (Ribeiro et al., 2017) uses a node similarity measure that explicitly builds on structural network properties. CNE, unlike the aforementioned methods, unifies the proximity in embeddings space and node similarity using a probabilistic measure. This allows CNE to find a more informative ML embedding.

The question of how to visualize networks on digital screens has been studied for a long time. Recently there has been an uplift in methods to embed networks in a 'small' number of dimensions, where small means small as compared to the number of nodes, yet typically much larger than two. These methods enable most machine learning methods to readily apply to tasks on networks, such as node classification or network partitioning. Popular methods include node2vec (Grover & Leskovec, 2016), where for example the default output dimensionality is 128. It is not designed for direct use in visualization, and typically one would fit a higher-dimensional embedding and then apply dimensionality reduction, such as PCA (Peason, 1901) or t-SNE (Maaten & Hinton, 2008) to visualize the data. CNE finds meaningful 2-d embeddings that can be visualized directly. Besides, CNE gives a visualization that conveys maximum information in addition to prior knowledge about the network.

## 5 CONCLUSIONS

The literature on NE has so far considered embeddings as tools that are used on their own. Yet, Euclidean embeddings are unable to accurately reflect certain kinds of network topologies, such that

this approach is inevitably limited. We proposed the notion of Conditional Network Embeddings (CNEs), which seeks an embedding of a network that maximally adds information with respect to certain given prior knowledge about the network. This prior knowledge can encode information about the network that cannot be represented well by means of an embedding.

We implemented this conceptually novel idea in a new algorithm based on a simple probabilistic model for the joint of the data and the network, which scales similarly to state-of-the-art NE approaches. The empirical evaluation of this algorithm confirms our intuition that the combination of structural prior knowledge and a Euclidean embedding is extremely powerful. This is confirmed empirically for both the tasks of link prediction and multi-label classification, where CNE outperforms a range of state-of-the-art baselines on a wide range of networks.

In our future work we intend to investigate other models implementing the idea of conditional NEs, alternative and more scalable optimization strategies, as well as the use of other types of structural information as prior knowledge on the network.

ACKNOWLEDGMENTS

The research leading to these results has received funding from the European Research Council under the European Union's Seventh Framework Programme (FP7/2007-2013) / ERC Grant Agreement no. 615517, from the FWO (project no. G091017N, G0F9816N), and from the European Union's Horizon 2020 research and innovation programme and the FWO under the Marie Sklodowska-Curie Grant Agreement no. 665501.

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

# Conditional Network Embeddings: Supplement

**Bo Kang, Jefrey Lijffijt & Tijl De Bie**
Department of Electronics and Information Systems (ELIS), IDLab
Ghent University
Ghent, Belgium
{firstname.lastname}@ugent.be

## 1 Derivation of the gradient

Denote the Euclidean distance between two points as $d_{ij} \triangleq ||\mathbf{x}_i - \mathbf{x}_j||_2$. The derivative of $d_{ij}$ with respect to embedding $\mathbf{x}_i$ of node $i$ reads:

$$\nabla_{\mathbf{x}_i} d_{ij} = \frac{\mathbf{x}_i - \mathbf{x}_j}{d_{ij}}$$

Then the derivative of the log posterior with respect to $\mathbf{x}_i$ is given by:

$$
\begin{aligned}
\nabla_{\mathbf{x}_i} \log\left(P(G|\mathbf{X})\right) =& \sum_{j:\{i,j\}\in E} \left( \frac{\partial \log\left(P(G|\mathbf{X})\right)}{\partial d_{ij}} + \frac{\partial \log\left(P(G|\mathbf{X})\right)}{\partial d_{ji}} \right) \nabla_{\mathbf{x}_i} d_{ij} \\
&+ \sum_{j:\{i,j\}\notin E} \left( \frac{\partial \log\left(P(G|\mathbf{X})\right)}{\partial d_{ij}} + \frac{\partial \log\left(P(G|\mathbf{X})\right)}{\partial d_{ji}} \right) \nabla_{\mathbf{x}_i} d_{ij} \\
=& \, 2 \sum_{j:\{i,j\}\in E} \frac{\partial \log\left(P(G|\mathbf{X})\right)}{\partial d_{ij}} \frac{\mathbf{x}_i - \mathbf{x}_j}{d_{ij}} + 2 \sum_{j:\{i,j\}\notin E} \frac{\partial \log\left(P(G|\mathbf{X})\right)}{\partial d_{ij}} \frac{\mathbf{x}_i - \mathbf{x}_j}{d_{ij}}
\end{aligned}
$$

Using shorthand notation $\mathcal{N}_{ij,\sigma_1} = \mathcal{N}_+\left(d_{ij}|\sigma_1^2\right)$ and $\mathcal{N}_{ij,\sigma_2} = \mathcal{N}_+\left(d_{ij}|\sigma_2^2\right)$, we can compute the partial derivative $\frac{\partial \log(P(G|\mathbf{X}))}{\partial d_{ij}}$ for $\{i,j\} \in E$ as:

$$
\begin{aligned}
\frac{\partial \log\left(P(G|\mathbf{X})\right)}{\partial d_{ij}} &= \frac{\partial}{\partial d_{ij}} \sum_{\{i,j\}\in E} \log\left(\mathcal{N}_{ij,\sigma_1}P_{ij}\right) - \log\left(\mathcal{N}_{ij,\sigma_1}P_{ij} + \mathcal{N}_{ij,\sigma_2}\left(1 - P_{ij}\right)\right) \\
&= \frac{\mathcal{N}_{ij,\sigma_1}P_{ij} \cdot \frac{-d_{ij}}{\sigma_1^2}}{\mathcal{N}_{ij,\sigma_1}P_{ij}} - \frac{\mathcal{N}_{ij,\sigma_1}P_{ij} \cdot \frac{-d_{ij}}{\sigma_1^2} + \mathcal{N}_{ij,\sigma_2}\left(1 - P_{ij}\right) \cdot \frac{-d_{ij}}{\sigma_2^2}}{\mathcal{N}_{ij,\sigma_1}P_{ij} + \mathcal{N}_{ij,\sigma_2}\left(1 - P_{ij}\right)} \\
&= -\frac{d_{ij}}{\sigma_1^2} + P(a_{ij}=1|\mathbf{X})\frac{d_{ij}}{\sigma_1^2} + P(a_{ij}=0|\mathbf{X})\frac{d_{ij}}{\sigma_2^2}
\end{aligned}
$$

Similarly, the partial derivative $\frac{\partial \log(P(G|\mathbf{X}))}{\partial d_{ij}}$ for $\{i,j\} \notin E$ reads:

$$\frac{\partial \log\left(P(G|\mathbf{X})\right)}{\partial d_{ij}} = -\frac{d_{ij}}{\sigma_2^2} + P(a_{ij}=1|\mathbf{X})\frac{d_{ij}}{\sigma_1^2} + P(a_{ij}=0|\mathbf{X})\frac{d_{ij}}{\sigma_2^2}.$$

The partial derivatives $\frac{\partial \mathcal{N}_{mn,\sigma}P_{mn}}{\partial d_{ij}}$ are nonzero only when $m = i$ and $n = j$, which gives the final gradient:

$$
\begin{aligned}
\nabla_{\mathbf{x}_i} \log\left(P(G|\mathbf{X})\right) = &\, 2 \sum_{j:\{i,j\}\in E} (\mathbf{x}_i - \mathbf{x}_j)P(a_{ij}=0|\mathbf{X})\left(\frac{1}{\sigma_2^2} - \frac{1}{\sigma_1^2}\right) \\
&+ 2 \sum_{j:\{i,j\}\notin E} (\mathbf{x}_i - \mathbf{x}_j)P(a_{ij}=1|\mathbf{X})\left(\frac{1}{\sigma_1^2} - \frac{1}{\sigma_2^2}\right)
\end{aligned}
\tag{1}
$$

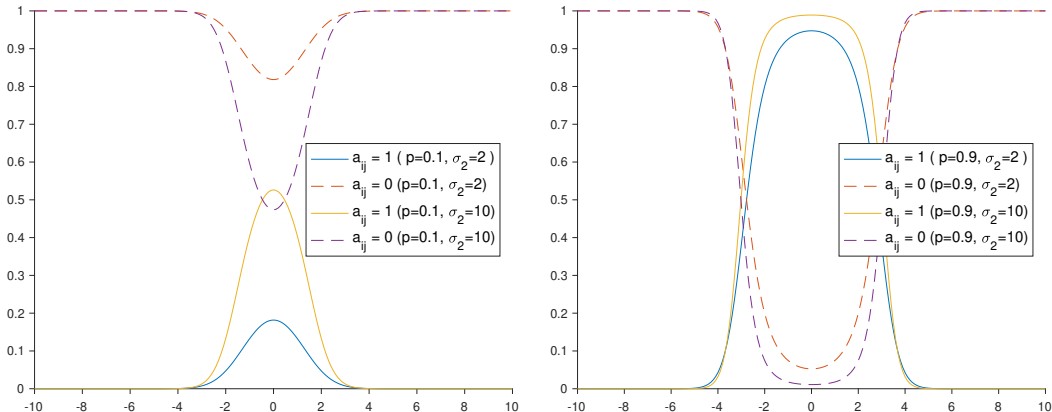

Figure 1: The posterior distribution $P(a_{ij} = 1|\mathbf{X})$ and $P(a_{ij} = 0|\mathbf{X})$ with different prior probability $P_{ij}$ and $\sigma_2$

## 2 Deriving the log probability of posterior $P(G|X)$

$$
\begin{aligned}
\log P(G|\mathbf{X}) &= \log \left( \prod_{\{i,j\}\in E} \frac{\mathcal{N}_{ij,\sigma_1} P_{ij}}{\mathcal{N}_{ij,\sigma_1} P_{ij} + \mathcal{N}_{ij,\sigma_2}(1-P_{ij})} \cdot \prod_{\{k,l\}\notin E} \frac{\mathcal{N}_{kl,\sigma_2}(1-P_{kl})}{\mathcal{N}_{kl,\sigma_1} P_{kl} + \mathcal{N}_{kl,\sigma_2}(1-P_{kl})} \right) \\
&= \log \left( \prod_{\{i,j\}\in E} \frac{1}{1 + \frac{\mathcal{N}_{ij,\sigma_2}(1-P_{ij})}{\mathcal{N}_{ij,\sigma_1} P_{ij}}} \cdot \prod_{\{k,l\}\notin E} \frac{1}{1 + \frac{\mathcal{N}_{kl,\sigma_1} P_{kl}}{\mathcal{N}_{kl,\sigma_2}(1-P_{kl})}} \right) \\
&= -\sum_{\{i,j\}\in E} \log \left( 1 + \frac{(2\pi\sigma_2^2)^{-1/2} \exp\left(-d_{ij}^2/(2\sigma_2^2)\right)(1-P_{ij})}{(2\pi\sigma_1^2)^{-1/2} \exp\left(-d_{ij}^2/(2\sigma_1^2)\right) P_{ij}} \right) \\
&\quad -\sum_{\{k,l\}\notin E} \log \left( 1 + \frac{(2\pi\sigma_1^2)^{-1/2} \exp\left(-d_{kl}^2/(2\sigma_1^2)\right) P_{kl}}{(2\pi\sigma_2^2)^{-1/2} \exp\left(-d_{kl}^2/(2\sigma_2^2)\right)(1-P_{kl})} \right) \\
&= -\sum_{\{i,j\}\in E} \log \left( 1 + \frac{\sigma_1}{\sigma_2} \frac{1-P_{ij}}{P_{ij}} \exp\left( \left( \frac{1}{\sigma_1^2} - \frac{1}{\sigma_2^2} \right) \frac{d_{ij}^2}{2} \right) \right) \\
&\quad -\sum_{\{k,l\}\notin E} \log \left( 1 + \frac{\sigma_2}{\sigma_1} \frac{P_{kl}}{1-P_{kl}} \exp\left( \left( \frac{1}{\sigma_2^2} - \frac{1}{\sigma_1^2} \right) \frac{d_{kl}^2}{2} \right) \right)
\end{aligned}
\tag{2}
$$

## 3 Effects of the $\sigma_1$ and $\sigma_2$ parameters

CNE seeks the embedding $\mathbf{X}$ that maximizes the likelihood $P(G|\mathbf{X})$ for given $G$. To understand the effect of parameter $\sigma_1$ and $\sigma_2$ we plot the posterior $P(a_{ij} = 1|\mathbf{X})$ as well as $P(a_{ij} = 0|\mathbf{X})$ in Figure 1. The plot shows a large $\sigma_2$ corresponds to more extreme minima of the objective function (Fig1a), thus results in stronger push and pulling effect in the optimization. Large link probability in the network prior further strengthen the pushing and pulling effects (Fig 1b). The flat area in Figure 1b ($\sigma_2 = 10$) allows connected nodes to keep some small distance from each other, and larger $\sigma_2$ also allows larger corrections to the prior probabilities (both Fig 1a and Fig 1b), but also makes the optimization problem harder.

## 4 Baseline methods used in experiments

We used the following baselines in the experiments:

- Deepwalk (Perozzi et al., 2014): This embedding algorithm learns embedding based on the similarities between nodes. The proximities are measured by random walks. The transition probability of walking from one node to all its neighbors are the same and are based on one-hop connectivity.

- LINE (Tang et al., 2015): Instead of random walks, this algorithm defines similarity between nodes based on first and second order adjacencies of the given network.

- node2vec (Grover & Leskovec, 2016): This is again based on random walks. In addition to its predecessors, it offers two parameters $p$, $q$ that interpolates the importance of BFS and DFS like random walk in the learning.

- metapath2vec++ (Dong et al., 2017): This approach is developed for heterogeneous NE, namely, the nodes belong to different node types. methapath2vec++ performs random walks by hopping from a node form one type to a node from another type. It also utilizes the node type information in the softmax based objective function.

- struc2vec (Ribeiro et al., 2017): The method first measures the structural information by computing pairwise similarity between nodes using a range of neighborhood sizes. This results in a multilayer weighted graph where the edge weights on the same layer are derived from the node similarity computed on one neighborhood size. Then the embedding is constructed by a random walk strategy that navigates the multilayer graph.

## 5 NETWORKS USED IN THE EXPERIMENTS

We used the following commonly used benchmark networks in the experiments:

- Facebook (Leskovec & Krevl, 2015): In this network, nodes are the users and links represent the friendships between the users. The network has 4,039 nodes and 88,234 links.

- arXiv ASTRO-PH (Leskovec & Krevl, 2015): In this network nodes represent authors of papers submitted to arXiv. The links represents the collaborations: two authors are connected if they co-authored at least one paper. The network has 18,722 nodes and 198,110 links.

- studentdb (Goethals et al., 2010): This is a snapshot of the student database from the University of Antwerp's Computer Science department. There are 403 nodes that belong to one of the following node types including: course, student, professor, program, track, contract, and room. There 3429 links that are the binary relationships between the nodes: student-in-track, student-in-program, student-in-contract, student-take-course, professor-teach-course, course-in-room. The database schema is given in Figure 2.

- Gowalla (Cho et al., 2011): This is a undirected location-based friendship network. The network has 196,591 nodes, 950,327 links.

- BlogCatalog (Zafarani & Liu, 2009): This social network contains nodes representing bloggers and links representing their relations with other bloggers. The labels are the bloggers' interests inferred from the meta data. The network has 10,312 nodes, 333,983 links, and 39 labels (used for multi-label classifications).

- Protein-Protein Interactions (PPI) (Breitkreutz et al., 2007): A subnetwork of the PPI network for Homo Sapiens. The subnetwork has 3,890 nodes, 76,584 links, and 50 labels.

- Wikipedia (Mahoney, 2011): This network contains nodes representing words and links representing the co-occurrence of words in Wikipedia pages. The labels represents the inferred Part-of-Speech tags (Toutanova et al., 2003). The network has 4,777 nodes, 184,812 links, and 40 different labels.

## 6 DETAILED RESULTS FOR MULTI-LABEL CLASSIFICATION

In the multi-label classification setting, each node is assigned one or more labels. For training, $50\%$ of the nodes and all their labels are used for training. The labels of the remaining nodes need to be predicted. We train CNE and baselines based on the full network. Then $50\%$ of the nodes are randomly selected to train a L2 regularized logistic regression classifier. The regularization strength

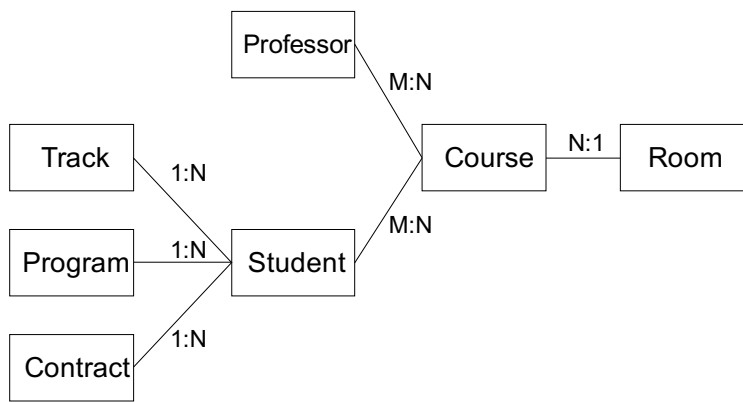

Figure 2: The entity relationship diagram of the studentdb dataset.

parameter of the classifier is trained with 10-fold cross-validation (CV) on the training data. We report the Macro-$F_1$ and Micro-$F_1$ based on the predictions. For the logistic regression classifier (sklearn, Pedregosa et al., 2011) we require every fold to have at least one positive and one negative label and we removed the labels that occur fewer than 10 times (number of folds in CV) in the data.

The detailed results of this approach based on logistic regression are shown in the upper half of Table 1. For CNE (written as CNE-LR to emphasize logistic regression was used for classifying), the embeddings are obtained with $d = 32$ and $k = 150$ (without optimizing). Somewhat surprisingly, CNE still performs in line with the state-of-the-art graph embedded methods, although without improving on them (on BlogCatalog, CNE performs third out of five methods, in PPI and Wikipedia it performs fourth out of five). This is surprising, given the fact that CNE yields embeddings that, by design, do not reflect certain information about the nodes that may be useful in classifying (here, their degree).

Multi-label classification can however be cast as a link prediction problem—a task we know CNE performs well at. To do this, we insert a node into the network corresponding to each of labels, and link the original nodes to the label nodes if they have that label. We can then employ link prediction, exactly as in the link prediction case (training on the full network, but with only $50\%$ of the edges between original nodes and label nodes, and the other half for testing), to do multi-label classification. For CNE, besides a degree prior, we can encode a 'block' prior which encodes the average connectivity between original nodes–original nodes, original nodes–labels, and labels–labels (which is zero, as labels are not connected to each other). Note that this approach means that also neighborhood-based link prediction methods can be used for multi-label classification.

The detailed results of this link prediction approach to multi-label classification are shown in the lower half of Table 1. CNE-LP (block+degree) (with LP to indicate it is based on link prediction) consistently outperforms all baselines on Macro-$F_1$, while on Micro-$F_1$ it is best on two datasets (BlogCatalog and PPI), and close second-best on one (Wikipedia). We note that while the benefit of this link prediction approach to multi-label classification is clear (and unsurprising) for CNE, there is no consistent benefit to other methods. This shows that the superior performance of CNE-LP for multi-label classification is not (or at least not exclusively) thanks to the link prediction approach, but at least in part also thanks to a more informative embedding when considered in combination with the prior.

## 7 RUNTIME EXPERIMENT

We compare the runtime (in second) of CNE with other baselines in this section. We use the parameters settings in link prediction task for all methods. Namely, for CNE, we set $d = 8$ (For arXiv $k = 16$ to reduce underfitting) and $k = 50$. We set stopping criterion of CNE $||\nabla_{\mathbf{X}}||_\infty < 10^{-2}$ or maxIter $< 250$ (whichever is met first). These stopping criteria yield embeddings with the same performance in link prediction tasks as reported in the paper. For other methods, we use the default setting as reported in their original paper. The hyper-parameters $p, q$ of node2vec are tuned using

Table 1: The $F_1$ scores for multi-label classification.

| Algorithm | BlogCatalog | | PPI | | Wikipedia | |
|---|---|---|---|---|---|---|
| | Macro-$F_1$ | Micro-$F_1$ | Macro-$F_1$ | Micro-$F_1$ | Macro-$F_1$ | Micro-$F_1$ |
| Multi-label classification using logistic regression (standard approach): | | | | | | |
| Deepwalk | 0.2544 | 0.3950 | 0.1795 | 0.2248 | 0.1872 | 0.4661 |
| LINE | 0.1495 | 0.2947 | 0.1547 | 0.2047 | 0.1721 | **0.5193** |
| node2vec | 0.2364 | 0.3880 | 0.1844 | 0.2353 | 0.1985 | 0.4746 |
| metapath2vec++ | 0.0351 | 0.1684 | 0.0337 | 0.0726 | 0.1031 | 0.3942 |
| struc2vec | 0.0493 | 0.1653 | 0.0669 | 0.0971 | 0.1124 | 0.4019 |
| CNE-LR (degree) | 0.1833 | 0.3376 | 0.1484 | 0.1952 | 0.1370 | 0.4339 |
| Multi-label classification through link prediction where labels are nodes: | | | | | | |
| Common Neigh. | 0.2115 | 0.2931 | 0.1792 | 0.1831 | 0.1212 | 0.3332 |
| Jaccard Sim. | 0.2157 | 0.1915 | 0.1799 | 0.1642 | 0.0552 | 0.0486 |
| Adamic Adar | 0.2301 | 0.3198 | 0.1698 | 0.1825 | 0.1035 | 0.3264 |
| Preferential Attach. | 0.2460 | 0.2084 | 0.2504 | 0.0953 | 0.2890 | 0.4454 |
| Deepwalk | 0.2372 | 0.2407 | 0.1848 | 0.1648 | 0.0876 | 0.0440 |
| LINE | 0.1599 | 0.2457 | 0.1052 | 0.1100 | 0.0976 | 0.2954 |
| node2vec | 0.2490 | 0.3462 | 0.2081 | 0.2069 | 0.1640 | 0.3057 |
| metapath2vec++ | 0.0633 | 0.1415 | 0.0571 | 0.0542 | 0.2021 | 0.3673 |
| struc2vec | 0.0644 | 0.1100 | 0.0631 | 0.0757 | 0.0905 | 0.3485 |
| CNE-LP (degree) | 0.2839 | 0.3929 | 0.2139 | 0.2303 | 0.1825 | 0.4407 |
| CNE-LP (block+degree) | **0.2935** | **0.4002** | **0.2639** | **0.2519** | **0.3374** | 0.4839 |

Table 2: The runtime (in seconds) of embedding methods. TimeOut means aborted after 24 hours.

| Algorithm | Facebook | PPI | arXiv | BlogCat. | Wikiped. | studentdb | Gowalla |
|---|---|---|---|---|---|---|---|
| Deepwalk | 120.78 | 116.09 | 714.68 | 344.72 | 138.89 | 8.34 | 5717.67 |
| LINE | 253.20 | 203.92 | 649.98 | 218.20 | 232.11 | 180.35 | 10988.71 |
| node2vec | 86.61 | 64.96 | 291.42 | 1054.73 | 288.32 | 6.04 | 5593.52 |
| metapath2vec++ | 130.78 | 39.59 | 274.60 | 332.19 | 78.14 | 3.50 | 333.29 |
| struc2vec | 2692.96 | 1105.41 | 54218.82 | 1356.67 | 1691.79 | 9245.23 | TimeOut |
| CNE (uniform) | 86.89 | 75.15 | 728.74 | 227.11 | 92.35 | 7.25 | 642.14 |
| CNE (degree) | 77.80 | 70.35 | 579.85 | 204.48 | 87.69 | 6.80 | 670.26 |
| CNE (block) | NA | NA | NA | NA | NA | 10.68 | NA |

cross validation. This experiment is performed with single process/thread on a desktop with CPU 2,7 GHz Intel Core i5 and RAM 16 GB 1600 MHz DDR3. Table 2 summarizes the runtime of all methods against all datasets we used in the paper. Over the seven datasets CNE is fastest in two cases, 12% slower than the fastest in one case (metapath2vec++), and approximately twice slower in the four other cases (also metapath2vec++).

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
