# OpenReview forum: "Conditional Network Embeddings"
_ICLR.cc/2019/Conference_

### Official Review · AnonReviewer1 · 2018-10-26
**Interesting idea, but some parts of the paper are not clear**

**Rating:** 5
**Confidence:** 4

**Review:**

This paper studied learning unsupervised node embeddings by considering the structural properties of networks. Experimental results on a few data sets prove the effective of the proposed approaches over existing state-of-the-art approaches for unsupervised node embeddings.

Strength:
- important problem and interesting idea
- the proposed approach seems to be effective according to the experiments
Weakness:
- some parts of the paper are quite unclear
- the complexity of the proposed algorithm seems to be very high
- the data sets used in the experiments are very small

Details:
-In the introduction, "it is in general impossible to find an embedding in R^d such that ...", why do we have to make v and v'(and u, and u') far from each other?
- In Equation (2), How is P_ij defined exactly, are they parameters? I am quite confused about this part
- In Equation (6), the posterior distribution should be P(X|G) since X is the latent variable to be inferred, right？
- In Table 2 and 3, how are the degree and block information leveraged into the model?

---

> ### Author Response · Authors · 2018-11-21
> **The computational as well as model complexity of CNE is low compared to baselines. We have added an experiment on a network of 200.000 nodes to support this.**
>
> We thank the reviewer for the thoughtful review.
>
> Responses:
> - We clarified the unclear parts, and will upload the revised version asap. (Also see below for specific responses.)
> - It is unclear to us if the reviewer thinks the computational complexity is high, or the mathematical complexity.
> With regards to mathematical complexity, we believe the model is actually rather simple (see also other reviews).
> Thus, we assume computational complexity is meant. Computational complexity is discussed in detail in the manuscript though, and is certainly not higher than competing methods (in part thanks to the low mathematical complexity of the model). See also next point.
> - The datasets we used are as large as the datasets used in other related work in the area. To demonstrate CNE's superior scalability, we included another network with around 200.000 nodes and around 1.000.000 edges (http://snap.stanford.edu/data/loc-Gowalla.html), run on a basic single CPU laptop. Again, CNE outperforms all other methods in accuracy by a wide margin, and is substantially faster as well. The results are included in the revised manuscript.
>
> Detailed comments:
>
> - "In the introduction, "it is in general impossible to find an embedding in R^d such that ...", [...]?"
> We apologize for having been a bit brief here, we will clarify this in the revision (uploaded asap). We meant to say that in network embedding methods that aim to model first-order proximity (where proximity in the embedding space implies a higher probability of being linked), this is a requirement (otherwise, proximity of v and v' would imply they are likely to be linked). Thus our argument only applies to such first-order proximity methods. Methods that aim to model second-order proximities (where proximity in the embedding space implies a greater overlap between the sets of adjacent nodes), however, are similarly vulnerable. For example, there can be a 50% overlap (which is highly significant in sparse networks) between the neighborhoods of nodes A and B, as well as between the neighborhoods of nodes B and C, but zero overlap between the neighborhoods of nodes A and C. This would mean that nodes A and B need to be embedded close to each other, nodes B and C as well, but nodes A and C distant from each other. The triangle inequality makes this hard. Finally, these are but examples of how a Euclidean embedding on its own lacks representational power. We believe that our empirical results also demonstrate this without having to refer to easy-to-identify problematic situations for pure embedding-based methods.
>
> - "In Equation (2), How is P_ij defined exactly [...]?"
> They are not parameters: they are numbers between 0 and 1 representing the prior probability of a link between nodes i and j (i.e. prior to seeing the embedding). These numbers are such that the prior knowledge of the types described are satisfied in expectation. In other words, they are implied and can be computed automatically and highly efficiently based on prior work, after one has decided on which prior knowledge to use.
> For details about how P_ij are fitted given such prior knowledge constraints, and how they can be represented efficiently, we have to refer to Adriaens et al. (2017) and van Leeuwen et al. (2016). We have however summarized the relevant aspects: the fact that all probabilities P_ij, although there are n^2 of them, can be represented using much fewer parameters, and the fact that they can be fitted highly efficiently (in our experiments, even on the largest networks, this always took only a tiny proportion of the total computation time). In the new version to be uploaded soon, this will be further clarified.
>
> - "In Equation (6), the posterior distribution should be P(X|G) [...]?"
> No, the equation is correct as stated. Footnote 2 warned the reader about this, as we know it is unusual. The posterior is a distribution for the network, such that finding the best embedding is indeed a maximum likelihood problem (not a maximum a posteriori problem), even though the likelihood function is computed as a posterior given a prior for the network and a conditional for the embedding given the network. We suspect that it is this unusual aspect of the CNE formulation that makes it original with respect to the state-of-the-art.
>
> - "In Table 2 and 3, how are [...]?"
> Equation (6) defines the posterior of the network given the embedding, which we maximize w.r.t. the embedding. It explicitly depends on the prior probabilities P_ij, which are computed based on the prior knowledge about the degrees or about the block density structure of the adjacency matrix. Thus, this information is brought into the model by considering the posterior distribution for the network, where the prior models the degrees of the nodes, or the block structure.

---

### Official Review · AnonReviewer3 · 2018-11-03
**The authors propose a generative model of networks via embeddings with the addition of a prior distribution over networks which facilitates learning more semantic embeddings. They use this model successfully in a variety of tasks.**

**Rating:** 6
**Confidence:** 3

**Review:**

The authors propose a generative model of networks by learning embeddings and pairing the embeddings with a prior distribution over networks. The idea is that the prior distribution may explain structure that the embeddings would not have to capture.

The motivation for doing this is that this structure is typically hard to model for network embeddings.
The authors propose a clean -if improper- prior on networks and proceed to perform maximum likelihood inference on it.
The experiments show that the approach works fine for link porediction and can be used for visualization.

Two points:
a) Why not try to do this with Variational inference? It should conceptually still work and be fast and potentially more robust.
b) The prior seems to be picked according to properties of the observed data and expressed in a product of constraints. This seems clunky, I would have been more impressed with a prior structure that ties in closer with the embeddings and requires less hand-engineering.

A key point of interest is the following: very exciting recent work (GraphRNN: Generating Realistic Graphs with Deep Auto-regressive Models by You et al ICML2018) has proposed neural generative models of networks with a high degree of fidelity and much less hand-picked features.  The work here tries to not learn a lot of these structures but impose them. Do the authors think that ultimately learning priors with models like GraphRNN might be more promising for certain applications?
The drawback in this model here is that ultimately networks are embedded, but not really generated during test time. A more predictive generative model that makes less hard assumptions on graph data would be interesting.

Update After rebuttal:
Given the authors' rebuttal to all reviews, I am upgrading my score to a 6. I still feel that more learning (as inGraphRNN) to build a fuller generative model of the graph would be interesting, but the authors make a strong case for the usefulness and practicality of their approach.

---

> ### Author Response · Authors · 2018-11-21
> **Variational inference is not needed. No hand-engineering is required.**
>
> We thank the reviewer for the thoughtful review.
>
> The reviewer points out that CNE works well for link prediction and visualization. We wish to point out that our experiments indicate CNE consistently outperforms the state-of-the-art not only for these tasks but also for multi-label classification.
>
> Our responses:
> a) Variational inference is useful in particular when the partition function is hard to compute, which is not the case here. So we believe it would be overkill in this case. In any case, it is not needed to achieve the performances CNE achieves at a very modest computational complexity and fast practical runtimes.
> b) No hand-engineering is needed to use CNE even when using more informative priors modeling node degrees and block structure.
> The degree of each node can simply be computed on the training set, so no hand-engineering is needed (just the choice to include it or not -- and it always better to include it).
> The block structure, on the other hand, will often be part of the data specification or meta-data of the nodes. For example, the network may be a multi-partite network representing a relational database, or it may be a company social network where the nodes are employees, and generic job titles are known for each of the employees (as attributes of the nodes). The entity types in the first example, and the job titles in the second example, would then define the blocks, and the density of the parts of the adjacency matrix between any two such blocks can again easily be computed on the network. Our method imposes no constraints on such blocks (e.g. they may even be partially overlapping). Again, all that is needed is choosing whether to use a block prior for any specified attribute (in the two examples: entity type, and node attribute). Again, empirically, including it always appears to be better, so one could even avoid having to make the choice.
>
> Inferring structural properties of the graph to be used in the prior (e.g. using GraphRNN), as we understand the reviewer suggests, certainly sounds potentially interesting. However, while it may improve accuracy, we do not believe that it adds value in reducing the amount of hand-engineering needed, as the amount of hand-engineering needed is very minimal already. The fact that CNE could be combined with such inferred structural properties increases its potential impact though, and this remark of the reviewer further underscores the need for methods such as CNE that can take such structural information into account.
>
> The boost in accuracy achieved by CNE, using a model that is arguably also a lot simpler than the state-of-the-art network embedding approaches, is thus achieved without any increased need for hand-engineering.

---

> > ### Comment · AnonReviewer3 · 2018-11-26
> > **Updated review available after rebuttal.**
> >
> > Thank you for your answer to his and the other reviews, it helped me position the work better as to its practicality and scope. Your comments/rebuttal have been reflected in my review.

---

### Official Review · AnonReviewer2 · 2018-11-04
**use prior distribution to constrain network embedding. novelty may not be high enough**

**Rating:** 4
**Confidence:** 4

**Review:**

The paper proposed to use a prior distribution to constraint the network embedding. The paper used very restricted Gaussian distributions for the formulation. The proposed approach should compared to other stronger methods such as graph convolution neural network/message passing neural networks/structure2vec.

---

> ### Author Response · Authors · 2018-11-21
> **A simple method that works well is arguably not worse, but better, than a more complex method that works well, and certainly than a more complex method that works worse**
>
> The reviewer feels that the proposed model is too simple, and suggests comparing against more complex models, suggesting a few in particular.
>
> Our response is twofold:
> - The accuracy of the proposed simple model exceeds the accuracy of far more complex models by a wide margin, and this consistently over a range of networks (all commonly used networks in this literature), against a range of baselines (all either commonly used baselines, or methods known as achieving state-of-the-art accuracies), and on two important tasks (link prediction and multi-label classification). We do not agree that its simplicity reduces its merit, we think it rather contributes to its merit.
> - We thank the reviewer for suggesting additional comparisons with specific more complex models, although we feel that calling these methods 'stronger' requires some clarification or support. Of the suggested methods, graph convolutional and message passing neural networks need attributed graphs as inputs, and are thus not applicable. We have in the meantime been able to include struc2vec in the evaluation, again showing superiority of CNE by a wide margin -- showing that it is maybe more complex but certainly not 'stronger' as in 'more accurate'. The paper will be updated very soon to include these results.
>
> Perhaps the reviewer is incredulous regarding this large increase in performance a method as 'simple' as CNE achieves w.r.t. the state-of-the-art. We believe that this is due to the conceptual advance made in CNE. In our opinion a conceptual advance that achieves a strong boost in accuracy without increasing complexity, is at least as valuable as a method that achieves the same boost in accuracy while also increasing complexity. Also note that all code is provided, and we invite the reviewer to replicate our experiments.

---

### Author Response · Authors · 2018-11-23
**New version uploaded with better explanations, and more experiments, showing superiority of CNE**

We have just uploaded a new version of the manuscript, which includes the changes announced in our responses to the reviewers.

In particular, some of the explanations have been reworded, and the experiments have been expanded in two directions:
- A larger network is now included (with around 200.000 nodes and around 1.000.000 edges).
- struc2vec is now included as an additional baseline.
These additional experiments demonstrate the scalability of CNE, and confirm that CNE achieves superior performance when compared against all baselines including struc2vec.

(To be able to add this additional material, we also had to shrink some other sections through reformulations.)

To summarize:
- CNE is mathematically elegant, and conceptually innovative.
- CNE achieves a consistently high performance, outperforming the state-of-the-art across a wide range of networks (all networks commonly used in the literature on this topic), against a wide range of baselines (including an additional applicable baseline suggested by a reviewer), and on standard tasks (including link prediction and multi-label classification) -- often by a large margin.
- CNE scales at least comparably to the most scalable baselines.

Regarding match to the scope of ICLR: CNE is not a deep learning approach, but it certainly is a representation learning method. We believe a 'shallow' approach that outperforms deep approaches for representation learning should be of great interest to ICLR.

We kindly invite the reviewers to take a second look at our paper, and hope they will be convinced.

---

### Author Response · Authors · 2018-11-26
**Please consider our responses**

Dear reviewers,

as we understand it, today is effectively the last day of the review period. We made a substantial effort to respond to your reviews and improve the paper. It would be great to hear your response.

Thank you very much, kind regards,

The authors

---

### Meta-Review · Area_Chair1 · 2018-12-19
**An intresting, novel approach to the network embedding problem on challenging graph structures, with uniformly better than state-of-art empirical results**

**Confidence:** 4
**Recommendation:** Accept (Poster)

**Metareview:**

The conditional network embedding approach proposed in the paper seems nice and novel, and consistently outperforms state-of-art on variety of datasets; scalability demonstration was added during rebuttals, as well as multiple other improvements; although  the reviewers did not respond by changing the scores, this paper with augmentations provided during the rebuttal appears to be a useful contribution  worthy of publishing at ICLR.